# Associations between Smoking and Smoking Cessation during Pregnancy and Newborn Metabolite Concentrations: Findings from PRAMS and INSPIRE Birth Cohorts

**DOI:** 10.3390/metabo13111163

**Published:** 2023-11-19

**Authors:** Brittney M. Snyder, Hui Nian, Angela M. Miller, Kelli K. Ryckman, Yinmei Li, Hilary A. Tindle, Lin Ammar, Abhismitha Ramesh, Zhouwen Liu, Tina V. Hartert, Pingsheng Wu

**Affiliations:** 1Department of Medicine, Vanderbilt University Medical Center, 2525 West End Avenue, Nashville, TN 37203, USAhilary.tindle@vumc.org (H.A.T.);; 2Department of Biostatistics, Vanderbilt University Medical Center, Nashville, TN 37203, USA; 3Division of Population Health Assessment, Tennessee Department of Health, Nashville, TN 37243, USA; 4Department of Epidemiology and Biostatistics, Indiana University School of Public Health—Bloomington, Bloomington, IN 47405, USA; 5Division of Family Health and Wellness, Tennessee Department of Health, Nashville, TN 37243, USA; yinmei.li@tn.gov; 6The Vanderbilt Center for Tobacco, Addiction and Lifestyle, Vanderbilt University Medical Center, Nashville, TN 37203, USA; 7Geriatric Research Education and Clinical Centers, Veterans Affairs Tennessee Valley Healthcare System, Nashville, TN 37212, USA; 8Vanderbilt University School of Medicine, Nashville, TN 37203, USA; lin.ammar.1@vumc.org; 9Department of Epidemiology, University of Iowa College of Public Health, Iowa City, IA 52242, USA; 10Department of Pediatrics, Vanderbilt University Medical Center, Nashville, TN 37203, USA

**Keywords:** prenatal smoking, third trimester, fetal metabolic programming, newborn metabolites

## Abstract

Newborn metabolite perturbations may identify potential biomarkers or mechanisms underlying adverse, smoking-related childhood health outcomes. We assessed associations between third-trimester smoking and newborn metabolite concentrations using the Tennessee Pregnancy Risk Assessment Monitoring System (PRAMS, 2009–2019) as the discovery cohort and INSPIRE (2012–2014) as the replication cohort. Children were linked to newborn screening metabolic data (33 metabolites). Third-trimester smoking was ascertained from birth certificates (PRAMS) and questionnaires (INSPIRE). Among 8600 and 1918 mother–child dyads in PRAMS and INSPIRE cohorts, 14% and 13% of women reported third-trimester smoking, respectively. Third-trimester smoking was associated with higher median concentrations of free carnitine (C0), glycine (GLY), and leucine (LEU) at birth (PRAMS: C0: adjusted fold change 1.11 [95% confidence interval (CI) 1.08, 1.14], GLY: 1.03 [95% CI 1.01, 1.04], LEU: 1.04 [95% CI 1.03, 1.06]; INSPIRE: C0: 1.08 [95% CI 1.02, 1.14], GLY: 1.05 [95% CI 1.01, 1.09], LEU: 1.05 [95% CI 1.01, 1.09]). Smoking cessation (vs. continued smoking) during pregnancy was associated with lower median metabolite concentrations, approaching levels observed in infants of non-smoking women. Findings suggest potential pathways underlying fetal metabolic programming due to in utero smoke exposure and a potential reversible relationship of cessation.

## 1. Introduction

Prenatal smoking is a leading modifiable risk factor for infant morbidity and mortality [1]. Despite overwhelming evidence of subsequent adverse health effects, smoking during pregnancy remains prevalent in the United States [2,3]. Smoking during the third trimester may have the greatest impact on adverse fetal health outcomes as this is a period of substantial fetal growth and stress on the placenta [1]. While several biological mechanisms have been proposed for observed relationships between prenatal smoking and adverse fetal health outcomes, including impaired fetal oxygenation [4] and metabolism [5,6], altered physiologic response and development [4], and toxin exposure [4], specific pathways underlying these associations remain unclear [6,7].

Metabolites are end-products of cellular responses to genetic and environmental changes [8,9]. Metabolism provides the body with energy and is essential for growth, development, movement, and reproduction [10]. Because the regulation of metabolites involved in these vital processes, such as free carnitine, acylcarnitines, and amino acids, is strictly maintained [11,12], perturbations in concentrations of these metabolites may point to pathways involved in disease pathogenesis [9,11]. Additionally, metabolites enriched in amino acid and lipid pathways have been shown to significantly differ between current and never smokers [13]. Therefore, assessing associations between smoking during the third trimester and concentrations of free carnitine, acylcarnitines, and amino acids at birth could provide important insights into targetable pathways and potential mechanisms underlying subsequent adverse fetal health outcomes.

We aimed to assess associations between smoking during the third trimester of pregnancy and newborn metabolite concentrations using Tennessee-specific data from the Centers for Disease Control and Prevention (CDC) Pregnancy Risk Assessment Monitoring System (PRAMS), the Infant Susceptibility to Pulmonary Infections and asthma following RSV Exposure (INSPIRE) birth cohort, and newborn screening (NBS) targeted, blood metabolic data. We additionally assessed whether smoking cessation during pregnancy is associated with newborn metabolite concentrations.

## 2. Materials and Methods

### 2.1. Study Design and Populations

We conducted a multi-cohort study of mother–child dyads of Tennessee residents who were selected to participate in the PRAMS survey from 2009–2019 (55% actually participated in the survey; Table 1) or were enrolled in the INSPIRE birth cohort from 2012–2014. The larger cohort, PRAMS, was used in the discovery phase, and the smaller cohort, INSPIRE, was used to replicate the findings. PRAMS study design methodology has been described previously [14]. Briefly, PRAMS is an ongoing, population-based, jurisdiction-specific, public health surveillance system for US women with a recent live birth [15]. Women were sampled from birth certificate records and contacted 2–6 months after delivery to collect information on their behaviors and experiences occurring before pregnancy, during pregnancy, and shortly after delivery. Demographic and clinical data were also available from linked birth certificates. INSPIRE is an ongoing, population-based birth cohort including term, non-low birth weight, healthy infants. Infants were enrolled shortly after birth from pediatric practices located in middle Tennessee. This cohort has been described previously [16].

For better harmonization with the INSPIRE birth cohort, we restricted to PRAMS children who were singleton, term birth (≥37 weeks gestation), and non-low birth weight (≥5 pounds). Children in both cohorts were linked to NBS targeted blood metabolic data. For both cohorts, children with missing NBS metabolic data due to refusal of NBS, metabolite concentrations outside the normal range, or incomplete linkage were excluded (Figure 1). We further excluded women with missing information on smoking during the third trimester of pregnancy. The study protocol (PRAMS and INSPIRE) and informed consent documents (INSPIRE only) were approved by the Vanderbilt University Medical Center and Tennessee Department of Health Institutional Review Boards.

### 2.2. Smoking Ascertainment

Our primary exposure was smoking during the third trimester of pregnancy (yes, no). For PRAMS, we used non-zero, average daily cigarette use during the last three months of pregnancy reported on the birth certificate as a surrogate measure for smoking during the third trimester of pregnancy (Appendix A). For INSPIRE, third-trimester cigarette use was ascertained from questionnaires administered after birth (infant aged ~2 months). Information on number of cigarettes smoked per day during this time was also collected for each cohort.

We further classified women based on their smoking status in the three months (PRAMS) or one year (INSPIRE) prior to pregnancy and during the third trimester of pregnancy as quitters (cigarette use prior to pregnancy but not in the third trimester of pregnancy), continued smokers (cigarette use both prior to pregnancy and in the third trimester of pregnancy), and non-smokers (no cigarette use prior to pregnancy and no use in the third trimester of pregnancy).

### 2.3. Newborn Screening Metabolic DATA Collection

Our primary outcomes were concentrations of 33 targeted metabolites at birth ascertained from the Tennessee NBS metabolic panel and provided by the Tennessee Department of Health NBS program. Tennessee NBS metabolic data include targeted measurement of free carnitine, 21 acylcarnitines, and 11 amino acids (Appendix A). The Tennessee NBS program only provided data for infants whose metabolite concentrations were within the normal range (i.e., screened negative for an inherited disorder) (Figure 1). This reduced the risk of potential participant identification and removed skewed metabolite concentrations due to inborn errors of metabolism [9,17].

### 2.4. Covariate Ascertainment

We selected covariates based on clinical relevance or published evidence of their association with prenatal smoking or infant metabolism. Maternal characteristics included maternal age at delivery, pre-pregnancy body mass index (BMI), maternal race and ethnicity, education, marital status, delivery method, type of health insurance, residence, pregnancy weight gain, pregnancy hypertension, and gestational diabetes. Infant characteristics included gender, birth weight, gestational age, ever breastfed, and birth year. For PRAMS, covariates were ascertained from linked birth certificates. For INSPIRE, all covariates, excluding maternal race and ethnicity, were ascertained from enrollment questionnaires. Maternal race and ethnicity were ascertained from maternal questionnaires administered when the child was aged 6 years.

### 2.5. Statistical Analysis

We compared maternal characteristics, infant characteristics, and metabolite concentrations between cohorts; individuals who did and did not smoke during the third trimester (within cohort); and non-smokers, quitters, and continued smokers (within cohort) using Mann–Whitney U, Kruskal–Wallis, or Pearson χ^2^ tests, as appropriate. For the primary analysis (Appendix A), we assessed associations between smoking during the third trimester and metabolite concentrations at birth using multivariable linear regression for metabolites with continuous values (n = 28 metabolites) or proportional odds regression for metabolites with <15 unique values (n = 5 metabolites) (Appendix A). Continuous metabolites were log-transformed to meet normality assumptions. We estimated the effects of smoking in terms of fold change in the median concentration for metabolites with continuous measurements or odds ratio of having higher metabolite concentrations for those with discrete measurements. We also calculated 95% confidence intervals (CIs) of the estimates. We used a directed acyclic graph (DAG) to identify the minimally sufficient covariate adjustment set required to reduce confounding when estimating the overall effect of smoking during the third trimester of pregnancy on metabolite concentrations at birth [18]. Based on the proposed DAG (Appendix A), models were adjusted for maternal age at delivery, maternal race and ethnicity, type of health insurance, maternal pre-pregnancy BMI, and ever breastfed. Metabolites identified in the discovery phase as significantly associated with smoking during the third trimester after covariate and *p*-value adjustment (false discovery rate adjusted for multiple testing) were then assessed in the replication phase.

For metabolites that remained statistically significant in the replication phase, we performed additional analyses in both cohorts, assessing potential dose-response associations between average number of daily cigarettes smoked during the third trimester and metabolite concentrations at birth using the same models outlined in the primary analysis, limiting to women who smoked during the third trimester (Appendix A). Restricted cubic splines were used for the number of daily cigarettes smoked during the third trimester, including knots located at the 10th, 50th, and 90th percentiles of the distribution of the number of daily cigarettes used.

To assess whether smoking cessation during pregnancy is associated with concentrations of metabolites that remained statistically significant in the replication phase, we conducted a separate set of analyses classifying women into non-smokers, quitters, and continued smokers (Appendix A). Associations between women’s patterns of smoking during pregnancy and newborn metabolite concentrations were assessed using the same models outlined in the primary analysis.

As sensitivity analyses, we performed multiple imputation within each cohort to handle missing covariates. All covariates described above, exposure variables (binary and categorical smoking status), and all metabolites were included in the imputation model. Imputation of missing covariate values was implemented using the *aregImpute* function in the R *Hmisc* package. Data analyses were performed using R software, version 4.1.0 (R Foundation for Statistical Computing, Vienna, Austria). *p*-values < 0.05 were considered statistically significant. Additional details on methodology can be found in the Appendix A.

## 3. Results

Our final study populations included 8600 and 1918 mother–child dyads in the discovery (PRAMS) and replication (INSPIRE) cohorts, respectively (Figure 1). Women who were selected for PRAMS were more racially and ethnically diverse than women in the INSPIRE cohort (PRAMS: 63% non-Hispanic White, 22% non-Hispanic Black, 9% Hispanic, 5% Other, <1% missing; INSPIRE: 68% non-Hispanic White, 19% non-Hispanic Black, 7% Hispanic, 6% Other, 0% missing) (Table 1). Compared to INSPIRE women, PRAMS women were more likely to reside in a rural setting, have lower levels of education, and were less likely to be married. PRAMS women were also more likely to have a vaginal delivery and less likely to breastfeed. Children selected for PRAMS were lower in birth weight compared to children enrolled in the INSPIRE cohort (Table 2).

During the third trimester of pregnancy, 14% and 13% of PRAMS and INSPIRE women reported smoking, respectively. Among women who smoked during the third trimester (PRAMS: n = 1218; INSPIRE: n = 240), the median number of cigarettes smoked per day was 10 (interquartile range [IQR] 5–10) in PRAMS and 6 (IQR 4–10) in INSPIRE (Appendix A). Most women who smoked during the third trimester of pregnancy also smoked both prior to pregnancy and during the first two trimesters of pregnancy (PRAMS: 95%, INSPIRE: 93%) (Appendix A). Comparison of maternal and infant characteristics by third-trimester smoking status and cohort are included in Appendix A. In both cohorts, women who smoked during the third trimester were more likely to be non-Hispanic White, younger, reside in a rural setting, have government insurance, be less educated, and have a cesarean section delivery compared to women who did not smoke during the third trimester. Women who smoked during the third trimester were also less likely to be married. Children with mothers who smoked during the third trimester had lower birth weights compared to those whose mothers did not smoke during the third trimester in INSPIRE but not PRAMS.

Metabolite concentrations at birth are provided by third-trimester smoking status and cohort in Appendix A. Smoking during the third trimester of pregnancy was associated with concentrations of several metabolites at birth in the discovery cohort (Figure 2 and Appendix A). When assessed in the replication cohort, third-trimester smoking remained statistically significantly associated with higher median concentrations of free carnitine (C0), glycine (GLY), and leucine (LEU) at birth compared to no third-trimester smoking (PRAMS: C0: adjusted fold change [exp(β)_adj_] 1.11 [95% confidence interval (CI) 1.08, 1.14], GLY: exp(β)_adj_ 1.03 [95% CI 1.01, 1.04], LEU: exp(β)_adj_ 1.04 [95% CI 1.03, 1.06]; INSPIRE: C0: exp(β)_adj_ 1.08 [95% CI 1.02, 1.14], GLY: exp(β)_adj_ 1.05 [95% CI 1.01, 1.09], LEU: exp(β)_adj_ 1.05 [95% CI 1.01, 1.09]).

In subsets of PRAMS (n = 1218) and INSPIRE (n = 240) women who smoked during the third trimester of pregnancy, further assessment of associations between the number of cigarettes women smoked per day and newborn concentrations of C0, GLY, and LEU did not show statistically significant and consistent dose-response relationships (Appendix A).

There were 79%, 7%, and 14% non-smokers, quitters, and continued smokers in the PRAMS cohort, respectively. The corresponding proportions of the groups in the INSPIRE cohort were 72%, 15%, and 12%. While continued smoking during pregnancy compared to non-smoking was significantly associated with higher median concentrations of C0, GLY, and LEU in both the discovery and replication cohorts (PRAMS: C0: exp(β)_adj_ 1.12 [95% CI 1.08, 1.14], GLY: exp(β)_adj_ 1.03 [95% CI 1.01, 1.05], LEU: exp(β)_adj_ 1.05 [95% CI 1.03, 1.06]; INSPIRE: C0: exp(β)_adj_ 1.08 [95% CI 1.03, 1.15], GLY: exp(β)_adj_ 1.05 [95% CI 1.01, 1.09], LEU: exp(β)_adj_ 1.04 [95% CI 1.01, 1.08]), quitting compared to continued smoking was associated with lower median concentrations of C0, GLY, and LEU in both the discovery and replication cohorts, approaching levels observed in infants of non-smoking women (Figure 3 and Appendix A). However, only the decrease in LEU was statistically significant in both cohorts (PRAMS: exp(β)_adj_ 0.97 [95% CI 0.95, 0.99], INSPIRE: exp(β)_adj_ 0.94 [95% CI 0.90, 0.98]) (Figure 3). In both cohorts, there were no statistically significant differences in the median concentrations of C0, GLY, and LEU between quitters and non-smokers. All the above results were unchanged after imposing multiple imputation strategies for missing data (Appendix A).

## 4. Discussion

We identified and replicated associations between smoking during the third trimester of pregnancy and higher concentrations of C0, GLY, and LEU at birth (within the normal range). We importantly showed that smoking cessation during pregnancy compared to continued smoking was associated with lower newborn C0, GLY, and LEU concentrations approaching those observed in newborns with non-smoking mothers.

While smoking has been shown to alter adult serum metabolite profiles [13], few studies have assessed the impact of in utero tobacco exposure on newborn metabolism. Our previous work failed to identify associations between prenatal smoking (defined as ever/never during pregnancy) and newborn metabolite concentrations [9]. This negative finding was likely due to our broad definition of prenatal smoking, which included a heterogeneous group of women who either continued smoking throughout pregnancy or quit smoking during pregnancy. As we have shown in the present study that 7% of PRAMS women and 15% of INSPIRE women quit smoking during pregnancy, and smoking cessation is associated with metabolite concentrations comparable to those observed in infants of non-smokers, a comparison between ever smoking with non-smoking drives the results in the prior study toward null.

Two additional studies have been performed in smaller populations without independent replication in separate cohorts. In a prospective cohort study of 40 mother–child pairs of full-term newborns, Rolle-Kampczyk et al. showed that maternal smoking was associated with both maternal and infant metabolic changes, often presented in opposite directions [19]. Using a Gaussian graphical model, the study found that cord blood concentrations of GLY and C0 were lower among infants of smoking mothers compared to infants of non-smoking mothers, a finding opposite to what we observed in the present study. In a separate study of 828 mother–child pairs [20], Cajachagua-Torres et al. assessed changes in neonatal metabolic profiles when exposed to tobacco smoke either continuously throughout pregnancy or in the first trimester only. Both continued tobacco exposure and in the first trimester only were associated in a dose-dependent manner with neonatal metabolite profile adaptations. Importantly, the authors showed that neonatal metabolic adaptation in response to tobacco exposure differed between those exposed in the first trimester only and those exposed continuously throughout pregnancy, a result consistent with what we reported in this study.

Tobacco smoke yields large quantities of reactive oxygen species (ROS) comprised of free and non-free radical oxygen intermediates, such as hydrogen peroxide and superoxide [21]. While ROS are naturally produced by cells through enzymatic processes, increased chronic levels of ROS can overwhelm the antioxidant system and lead to oxidative stress and damage [21]. During pregnancy, oxidative damage caused by tobacco smoke can affect not only the lungs of the mother but also placental tissue [22]. Placental tissue and amniotic fluid collected from mothers who smoke have decreased total antioxidant capacity, an oxidative-dominant shift in the oxidative/antioxidative balance, and increased levels of oxidative markers [22,23,24]. There is also evidence to suggest that oxidative damage to placental tissue transfers to the infant, as higher levels of oxidative stress markers have been observed in infants with mothers who smoked during pregnancy compared to infants with mothers who did not smoke during pregnancy [25,26,27]. Oxidative stress can induce apoptosis and cellular senescence and activate the inflammatory response pathway [21]. Additionally, oxidative damage caused by tobacco inhalation can lead to maternal lipid peroxidation (i.e., fatty acid degradation) and protein modifications [21]. Increased apoptosis and markers of lipid peroxidation have also been observed in the placental tissues of mothers who smoke [22,28].

The observed associations between smoking during the third trimester of pregnancy and higher concentrations of C0, GLY, and LEU at birth observed in this study may be due to fetal compensatory mechanisms aimed at reducing oxidative damage, apoptosis, inflammation, and lipid peroxidation induced by smoke exposure. C0 has been shown to be a potent antioxidant due to its capacity to scavenge free radicals and protect the antioxidant defense system from peroxidative damage [29]. Additionally, C0 reduced ROS formation, lipid peroxidation, and mitochondrial dysfunction in experimental models [29,30] and decreased apoptosis while promoting cellular proliferation through its stimulating effect on mitochondria and inhibition of pro-inflammatory cytokines [31,32,33]. GLY prevents ROS formation through multiple pathways, including inhibition of macrophage activation (and subsequent cytokine production and transcription factor activation) and minimization of antioxidant enzyme impairment [34]. GLY has also been shown to suppress inflammatory cytokine formation, protect against cell injury through inhibition of degradative enzyme activation, and prevent tissue hypoxia through improved microcirculation [34]. Branched-chain amino acids (BCAAs), including LEU, are essential to the immune system, as they are used in the synthesis and fueling of immune cells [35]. LEU regulates the immune system through the mTOR pathway, which regulates innate and adaptive immune responses and promotes differentiation, activation, and function of T cells, B cells, and antigen-presenting cells [35]. In addition to its immunomodulatory role, there is some experimental evidence to suggest LEU’s antioxidative role, including its ability to increase total antioxidant capacity and decrease plasma free radical concentrations [36,37,38,39]. Catabolism of BCAAs, mainly LEU, is used in the synthesis of glutamate [40]. Glutamate is an essential substrate in glutathione synthesis, which is the main non-enzymatic intracellular antioxidant [41].

Our study has many strengths, including our large sample sizes and replication of findings in an independent population. Variation in metabolite concentrations at birth due to gestational age and birth weight [42] was minimized through the restriction of our populations to term birth, non-low birth weight children. As the collection and measurement of NBS samples and data are standardized [43,44], the risk of measurement bias was minimal. In addition, we employed a rigorous *a priori* statistical analysis plan, including adjustment for multiple comparisons and multiple imputation in sensitivity analyses, to reduce multiple testings and account for missing data.

Our study also has limitations. The targeted metabolites studied were limited to those quantified on the Tennessee newborn screening panel. We were further limited to a population with concentrations within the normal range. However, the metabolites studied are involved in vital processes and tightly regulated [11,12], and as such, perturbations in concentrations of these metabolites, even mild, may point to pathways involved in disease pathogenesis [9,11]. While smoking may contribute to concentrations outside of the normal range, clinically defined abnormal values are most likely due to a genetic disorder (i.e., inborn errors of metabolism). Additional biologic factors that act to modify metabolite levels, such as levels of adipose tissue [45] and somatomedins/insulin-like growth factor [46], may play a role in the observed associations. While we were unable to adjust for these biological factors in the present study, future studies should consider the potential role of these factors in the relationships between prenatal smoking and newborn metabolite concentrations. As this was an observational study, we were unable to determine if the observed perturbations in C0, GLY, and LEU are a direct result of the detrimental effects of smoking, as previous literature suggests, or are merely biomarkers of smoking. Future experimental work exploring the causal relationship is needed. For PRAMs, we used the last three months of pregnancy as a surrogate measure for the third trimester of pregnancy. Although these measures are not necessarily equal, our restriction to a term birth population increased the probability that these measures aligned. We defined quitting, continued smoking, and non-smoking during pregnancy based on women’s pre-pregnancy and third-trimester smoking status.

While continued smokers and non-smokers more likely to perform consistently during pregnancy, women classified as quitters may have started quitting at different stages of pregnancy (see Appendix A), and we did not consider the exact time of quitting in our analysis. We relied on self-reported smoking to determine women’s smoking status during pregnancy, which, as past studies suggest, is fairly accurate and representative of an individual’s smoking pattern in pregnancy [47,48,49,50]. For infants with missing maternal race and ethnicity in INSPIRE, we used infant race and ethnicity as a proxy variable. Although this is an imperfect measure, the kappa value was large (0.66), and the use of this surrogate measure is unlikely to have impacted the results. 

## 5. Conclusions

In this multi-cohort study, we identified and replicated associations between smoking during the third trimester of pregnancy and higher concentrations of C0, GLY, and LEU at birth. We further showed that smoking cessation during pregnancy is associated with lower concentrations of these metabolites approaching levels observed among infants of non-smokers, suggesting a potential reversible relationship of cessation. This study provides insights into potential pathways underlying fetal metabolic programming due to in utero smoke exposure. Future studies exploring whether elevated C0, GLY, and LEU at birth increase the risk of adverse infant outcomes may lead to interventions minimizing the adverse effects of maternal smoking.

## Figures and Tables

**Figure 1 metabolites-13-01163-f001:**
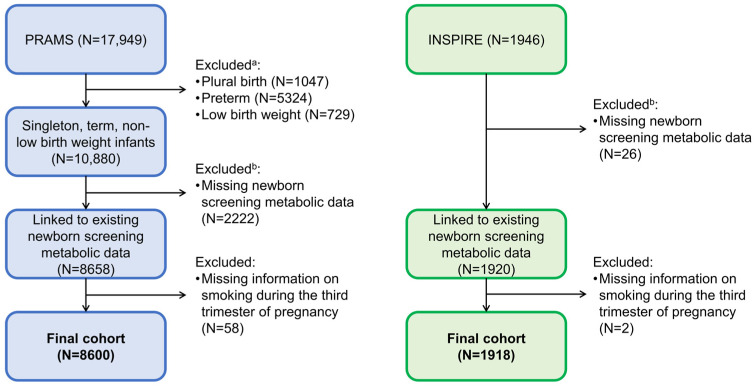
Flow diagram of study populations. ^a^ Not mutually exclusive. ^b^ Newborn screening metabolic data may have been missing due to refusal of newborn screening, metabolite concentrations outside the normal range, or incomplete linkage.

**Figure 2 metabolites-13-01163-f002:**
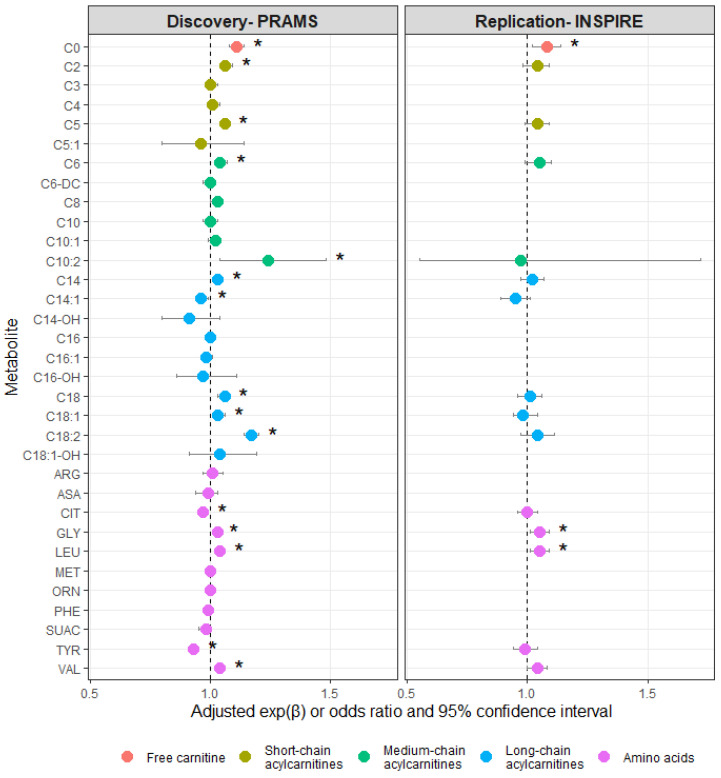
Associations between smoking during the third trimester of pregnancy and metabolite concentrations at birth by cohort. Fold change increases in the medians (exp[β]) were calculated for metabolites with continuous distributions (log-transformed) using linear regression. Odds ratios were calculated for metabolites with ordinal distributions (C5:1, C10:2, C14-OH, C16-OH, and C18:1-OH) using proportional odds regression. Regression models were adjusted for maternal age at delivery, maternal race and ethnicity, type of health insurance, maternal pre-pregnancy BMI, and ever breastfed. Associations were only explored in the replication cohort (INSPIRE) if statistically significant in the discovery cohort (PRAMS). * Adjusted *p*-value significant at α < 0.05. Discovery cohort *p*-values corrected for false discovery rate.

**Figure 3 metabolites-13-01163-f003:**
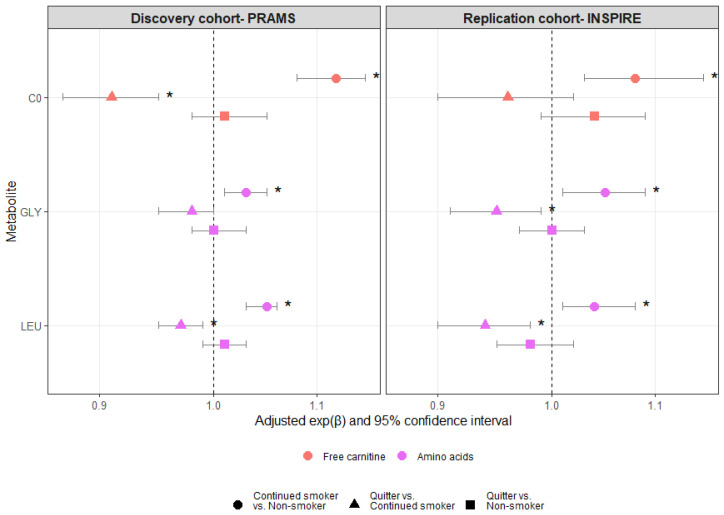
Associations between patterns of smoking during pregnancy and metabolite concentrations at birth by cohort. Fold change increases in the medians (exp[β]) were calculated for metabolites with continuous distributions (log-transformed) using linear regression. Regression models were adjusted for maternal age at delivery, maternal race and ethnicity, type of health insurance, maternal pre-pregnancy BMI, and ever breastfed. * Adjusted *p*-value significant at α < 0.05.

**Table 1 metabolites-13-01163-t001:** Maternal characteristics of the study populations with linked newborn screening metabolic data.

Maternal Characteristic	PRAMS	INSPIRE	*p*-Value ^a^
Sample size, n (%)	8600	1918	
Race and ethnicity, n (%)			<0.001 *
Non-Hispanic White	5448 (63)	1312 (68)	
Non-Hispanic Black	1928 (22)	366 (19)	
Hispanic	797 (9)	130 (7)	
Other ^b^	410 (5)	110 (6)	
Missing, n (%)	17 (0)	0 (0)	
Education (years), n (%)			<0.001 *
<12	1480 (17)	152 (8)	
12	2550 (30)	523 (27)	
13–15	2549 (30)	573 (30)	
≥16	1995 (23)	669 (35)	
Missing, n (%)	26 (0)	1 (0)	
Marital status, n (%)			<0.001 *
Married	4538 (53)	1102 (57)	
Other ^c^	4062 (47)	816 (43)	
Missing, n (%)	0 (0)	0 (0)	
Age at delivery (years), n (%)			<0.001 *
<20	881 (10)	149 (8)	
20–24	2314 (27)	548 (29)	
25–29	2533 (29)	555 (29)	
30–34	1887 (22)	487 (25)	
≥35	985 (11)	179 (9)	
Missing, n (%)	0 (0)	0 (0)	
Delivery method, n (%)			0.006 *
Vaginal	6178 (72)	1318 (69)	
Cesarean section	2422 (28)	600 (31)	
Missing, n (%)	0 (0)	0 (0)	
Insurance, n (%)			<0.001 *
Government	4525 (53)	1041 (54)	
Private	3546 (41)	855 (45)	
Other ^d^	217 (3)	20 (1)	
Missing, n (%)	312 (4)	2 (0)	
Residence, n (%)			<0.001 *
Urban	3810 (44)	1448 (75)	
Rural	3953 (46)	453 (24)	
Missing, n (%)	837 (10)	17 (1)	
Pre-pregnancy BMI, n (%)			0.007 *
<18.5	460 (5)	66 (3)	
18.5–24.9	3843 (45)	846 (44)	
25.0–29.9	2011 (23)	468 (24)	
≥30.0	2067 (24)	466 (24)	
Missing, n (%)	219 (3)	72 (4)	
Pregnancy weight gain (kgs), median (IQR)	14 (10–18)	15 (10–19)	<0.001 *
Pregnancy weight gain (lbs), median (IQR)	31 (21–40)	32 (23–42)	<0.001 *
Missing, n (%)	397 (5)	86 (4)	
Pregnancy hypertension, n (%)	651 (8)	153 (8)	<0.001 *
Missing, n (%)	0 (0)	755 (39)	
Gestational diabetes, n (%)	458 (5)	126 (7)	0.03 *
Missing, n (%)	0 (0)	0 (0)	
Participated in PRAMS survey ^e^	4718 (55)	N/A	N/A
Missing, n (%)	0 (0)		

BMI—body mass index; IQR—interquartile range; N/A—not applicable. ^a^
*p*-values calculated using Mann–Whitney U or Pearson χ^2^, as appropriate. ^b^ ‘Other’ category included Non-Hispanic Asian, Non-Hispanic American Indian, Non-Hispanic Chinese, Non-Hispanic Japanese, Non-Hispanic Filipino, Non-Hispanic mixed race, and Non-Hispanic other race for PRAMS and Non-Hispanic Asian, Non-Hispanic Hawaiian, Non-Hispanic multiple race, and Non-Hispanic Native American for INSPIRE. ^c^ ‘Other’ category included single and separated/divorced for INSPIRE. ^d^ ‘Other’ category included self-pay, Indian Health System, and ‘other’ categories for PRAMS and self-pay and ‘other’ categories for INSPIRE. ^e^ Only a proportion of women selected to participate in the PRAMS survey actually participated. Our study population included women selected to participate, not limited to those who participated. * Statistically significant at α < 0.05.

**Table 2 metabolites-13-01163-t002:** Infant characteristics of the study populations with linked newborn screening metabolic data.

Infant Characteristic	PRAMS	INSPIRE	*p*-Value ^a^
Sample size, n (%)	8600	1918	
Gender, n (%)			0.009 *
Female	4369 (51)	911 (47)	
Male	4231 (49)	1007 (53)	
Missing, n (%)	0 (0)	0 (0)	
Birth weight (grams), median (IQR)	3260 (2835–3572)	3405 (3120–3740)	<0.001 *
Missing, n (%)	0 (0)	0 (0)	
Gestational age (weeks), median (IQR)	39 (38–39)	39 (39–40)	<0.001 *
Missing, n (%)	0 (0)	0 (0)	
Ever breastfed, n (%)	6130 (71)	1469 (77)	0.01 *
Missing, n (%)	286 (3)	0 (0)	
Birth year, n (%)			<0.001 *
2009	566 (7)	N/A	
2010	1092 (13)	N/A	
2011	1124 (13)	N/A	
2012	699 (8)	844 (44)	
2013	644 (7)	1074 (56)	
2014	169 (2)	N/A	
2015	439 (5)	N/A	
2016	1065 (12)	N/A	
2017	1135 (13)	N/A	
2018	1064 (12)	N/A	
2019	603 (7)	N/A	
Missing, n (%)	0 (0)	0 (0)	

IQR—interquartile range; N/A—not applicable. ^a^
*p*-values calculated using Mann–Whitney U or Pearson χ^2^, as appropriate. * Statistically significant at α < 0.05.

## Data Availability

The datasets utilized in the present study include confidential newborn screening data that cannot be shared without approval from the Tennessee Department of Health Institutional Review Board. INSPIRE data presented in this study are available on request from the corresponding author. Requests for access to PRAMS data can be directed to the Tennessee Department of Health.

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
