# Peer review of "Associations between Smoking and Smoking Cessation during Pregnancy and Newborn Metabolite Concentrations: Findings from PRAMS and INSPIRE Birth Cohorts"

_metabolites, 2023, doi:10.3390/metabo13111163_

Round 1

Reviewer 1 Report

Comments and Suggestions for Authors

        This study examined the differences in new born metabolites in relation to maternal smoking in the third trimester of pregnancy. Their results suggest that maternal smoking was associated with changes in free carnitine, leucine, and glycine.

1. Introduction. Line 63-64. It may be helpful to provide some explanation why these metabolites were selected? Any prior evidence that they may be more susceptible to change due to tobacco exposure?

2. Line 108. How were normal ranges defined? What was the rationale of exclusion for those that were outside the normal range, don’t these “abnormal” values suggest potential alterations due to tobacco exposure or other disorders?

3. Were any maternal metabolite data available to assess association?

Reviewer 2 Report

Comments and Suggestions for Authors

This is an excellent paper. It's main force lies in great number of examinees- pregnant women. Minor suggestion: please use SI measures: kg along with the lbs because of the rest of the world reading this article. Please correct.

Reviewer 3 Report

Comments and Suggestions for Authors

This study is significant because it provides insights into potential pathways underlying fetal metabolic programming due to in-utero smoke exposure. The methods and analysis are appropriate.

Minor comment

1.       Line numbers 72-73

As stated in the study limitations, "Future experimental work exploring the causal relationship is needed." This study did not demonstrate an "effect" of smoking cessation during pregnancy. Please revise the manuscript to be more accurate so that the study shows a correlation, not a causal relationship.

2.       Figure 2.

It would be more precise to add an axis label of 1.0 to the horizontal axis of Figure 2. as in Figure 3.

Reviewer 4 Report

Comments and Suggestions for Authors

Report on manuscript

Associations between smoking and smoking cessation during pregnancy and newborn metabolite concentrations: Findings from PRAMS and INSPIRE birth cohorts

·       Overview of manuscript

The authors evaluated the associations between third trimester smoking and newborn metabolite concentrations. They observed that smoking cessation vs. continued smoking during pregnancy was associated with lower median metabolite concentrations.

·       Comments on text

1.     New contribution

The authors confirm potential pathways underlying fetal metabolic programming due to in utero smoke exposure, suggesting a reversible effect of cessation.

2.      English

The English in this paper is good.

Comments

Very interesting study and well performed analysis. The authors have provided enough supporting information for their findings.

The authors are pleased to consider the following suggestion: that would be great if the authors can have a brief introduction on adipose tissue, somatomedins or insulin growth factor, and birth weight.

·       Recommendation

After all the revisions, I can recommend acceptance of this paper.
